# Using Human-Centered Design to Bridge Zero-Dose Vaccine Gap: A Case Study of Ilala District in Tanzania

**DOI:** 10.3390/vaccines13010038

**Published:** 2025-01-06

**Authors:** Simon Martin Nzilibili, Hellen Maziku, Awet Araya, Ruthbetha Kateule, Millenium Anthony Malamla, Suna Salum, Furaha Kyesi, Lotalis Gadau, Tumaini Menson Haonga, Florian Tinuga, Rashid Mfaume, Zaitun Hamza, Georgina Joachim, Alice Geofrey Mwiru, Alex Benson, Oscar Kapela, Ona Machangu, Norman Jonas, Ntuli Kapologwe

**Affiliations:** 1Health Promotion Section, Ministry of Health, Dodoma P.O. Box 743, Tanzania; tumeha@yahoo.com (T.M.H.); kaperaoscar20@gmail.com (O.K.); onamach@gmail.com (O.M.); normanjonasmd@gmail.com (N.J.); 2Department of Computer Science and Engineering, College of ICT, University of Dar es Salaam, Dar es Salaam P.O. Box 35091, Tanzania; nahelna@gmail.com (H.M.); rnackyy@gmail.com (R.K.); milleniummalamla@gmail.com (M.A.M.); sunnasalum382@gmail.com (S.S.); 3Social Behavior Change UNICEF, Tanzania Country Office, Dar es Salaam P.O. Box 4076, Tanzania; aaraya@unicef.org (A.A.); amwiru@unicef.org (A.G.M.); 4Immunization and Vaccine Development Program, Ministry of Health, Dodoma P.O. Box 743, Tanzania; furahakyesi@hotmail.com (F.K.); lgadawu@gmail.com (L.G.); floriantinuga@yahoo.com (F.T.); joachimgeorgina@gmail.com (G.J.); 5Directorate of Health, Social Welfare and Nutrition Services at the President’s Office, Regional Administration and Local Government, Dodoma P.O. Box 1923, Tanzania; srashid302@gmail.com; 6Council Health Management Team, Dar es Salaam City Council, Dar es Salaam P.O. Box 77742, Tanzania; drzelha@gmail.com; 7School of Journalism and Mass Communication, University of Dar es Salaam, Dar es Salaam P.O. Box 35091, Tanzania; benzanda@gmail.com; 8Department of Preventive Services, Ministry of Health, Dodoma P.O. Box 743, Tanzania; nkapologwe2002@gmail.com

**Keywords:** zero dose, under-five vaccination, human centered design, advocacy, community-centric intervention

## Abstract

**Background:** Immunization plays a substantial role in reducing the under-five mortality rate. However, Tanzania still has a significant number of zero-dose and under-vaccinated children and was ranked among the top ten African countries with the highest numbers of zero-dose children in 2022. The human-centered design (HCD) approach is more ethical and effective at addressing public health challenges in complex sociocultural settings. This study aimed to use the HCD approach to aid in identifying, prioritizing, and implementing community-centric interventions in Tanzania, particularly in the Ilala District of Dar es Salaam, to increase vaccine demand and close the zero-dose gap by at least 50%. **Methods:** The study involved co-creation workshops with 483 participants to identify, design, and test solutions. The study followed the UNICEF Journey to Health and Immunization framework to identify barriers and enablers influencing stakeholders in adopting and sustaining health- and immunization-related actions. **Results:** The study identified the causes of under-five defaulting and the zero-dose gap, i.e., the inadequate support of local community leaders in under-five vaccination sensitization and surveillance; poor infrastructure to new settlement areas; hesitancy and unwillingness of parents/guardians; absence of house numbers; limited/time-constrained availability of resources to facilitate mobile immunization services, etc. The participants were able to come up with 309 ideas, which were refined through multiple iterations using the impact–-effort matrix and skimmed down to three (3) solutions: (i) having health facilities to notify and alert local leaders about vaccination dates; (ii) using parents, kids, and grownups who got vaccinated to influence others; (iii) using local government leaders and house representatives for vaccine advocacy. Of these, the solution involving local government leaders and house representatives for vaccine advocacy was implemented. An advocacy strategy was used to enhance the collaboration of the District Commissioner, Council leaders, and community leaders. A home-to-home interpersonal sensitization approach accompanied by the household delivery of vaccination services was employed. The findings reveal that the HCD framework was impactful in increasing collaborations/cooperation with local government leaders and community ownership of the under-five vaccination initiative. As a result, 67,145 houses, equal to 104%, were reached, surpassing the initial target of 64,800 houses, and 131,088 families, equal to 83% of the targeted 156,995 households, were sensitized through a home-to-home campaign approach. This study demonstrates the effectiveness of the approach. Researchers and practitioners are encouraged to adopt the HCD approach when addressing public health challenges, especially in complex sociocultural settings.

## 1. Introduction

The global mortality rate among children under five years of age declined by 59%, from 93 deaths per 1000 live births in 1990 to 37 deaths per 1000 live births in 2022 [1]. This significant reduction has been primarily attributed to the provision of childhood vaccinations, contributing to achieving the fourth United Nations Millennium Development Goal, which aimed to promote healthy lives for children and eliminate preventable deaths among newborns and young children [2]. Although substantial progress has been made in improving child survival, the magnitude of under-five mortality remains a significant burden in sub-Saharan Africa and South Asia [1,3]. Hence, continued efforts to reduce child mortality further are imperative.

In addition to the apparent health benefits, vaccination of under-five children is highly considered the most cost-effective, reliable, and robust public health intervention for curbing morbidity and mortality associated with preventable diseases [2,4]. Globally, immunization coverage has vastly expanded to cover all countries with the services available and accessible to rural and urban citizens of every nation [5]. In recent years, more efforts have been directed toward strict adherence to the vaccination schedule to immunize the under-five with vaccines such as Diphtheria–Tetanus, Measles–Rubella, and Polio [6]. The World Health Organization, in collaboration with governments of other countries, has implemented different strategies to upscale vaccine uptake among children [2]. As a result, these strategies increased the delivery of vaccination services and raised citizens’ awareness of the vaccination benefits to under-fives. Yet, millions of children in Africa are still deprived of life-saving vaccinations, accounting for nearly half of the world’s unvaccinated and under-vaccinated children [7].

Sub-Saharan African countries, particularly Tanzania, have recorded remarkable successes in nationwide child immunization coverage, attributed to vaccine availability, effective vaccine delivery mechanisms, and community sensitization and vaccination promotions [8]. Despite these accomplishments, significant challenges remain, including high rates of under-five mortality [9] and a considerable number of zero-dose and under-vaccinated children due to vaccine hesitancy [10]. In Tanzania, one in four children are not fully vaccinated. Many regions fail to meet the 90% coverage target set by the Reaching Every District (RED) strategy [10], and according to the WHO (2023), Tanzania, Angola, and Cameroon ranked among the top ten African countries with the highest numbers of zero-dose children in 2022 [11].

Various factors contribute to vaccine hesitancy, including trust, occupational status, education, marital status, wealth, place of residence, media exposure, and access to antenatal care (ANC) and postnatal care (PNC) [12,13,14]. These factors are shaped by social, economic, and cultural dynamics influencing parental vaccine awareness. For instance, the lack of trust in vaccines and healthcare practitioners is often fueled by misinformation, cultural beliefs, and past negative experiences with vaccination [13,15]. This distrust is a key factor contributing to the lack of confidence in the efficacy and safety of vaccines. Both occupations and education influence the extent to which an individual is exposed to vaccine-related information. For example, healthcare workers are exposed to detailed vaccine information, which can lead to either high acceptance or hesitancy [12,14]. Parental marital status can also influence vaccination hesitancy, as making a decision about whether to vaccinate a child often requires agreement between both parents [12,16]. Individuals with stable and higher financial resources generally have better access to healthcare services and vaccine-related information, making them more likely to accept vaccines compared to those with limited financial resources and influenced by other beliefs [14,17]. There is a critical need for effective interventions to overcome these barriers and ensure that children from hesitant families receive vaccinations. According to the WHO, interventions targeting vaccine hesitancy should be specifically tailored to the concerns of the respective target population to maximize their effectiveness [18].

Human-centered design (HCD) is a more ethical and practical approach to addressing public health challenges in complex sociocultural settings [19]. As conceptualized by the IDEO (2015), HCD emphasizes creating interventions tailored to the target population’s needs, preferences, and experiences [20]. This approach prioritizes collaboration with end-user communities, engaging deeply with the impacted individuals and groups. By fostering this co-creation process, HCD ensures that solutions are effective, culturally, and contextually relevant, making them more sustainable and likely to succeed [21]. Despite the substantial contributions of HCD in addressing complex problems in public health, there is a lack of empirical evidence describing the implementation of an entire HCD project [19]. Most studies are centered on one or two aspects of HCD, i.e., stakeholder engagement [22], ideation [23], or prototyping [24]. To address this gap, this study described the use of the HCD approach in identifying, prioritizing, and implementing community-centered interventions to increase vaccine demand and close the zero-dose gap in Tanzania, focusing on the Ilala District in Dar es Salaam. Ilala is the leading district in Dar es Salaam, with a high number of zero-dose and under-vaccinated children [25]. The intention was to reduce the number of children with zero doses of Diphtheria–Tetanus–Pertussis (DTP) and Measles–Rubella (MR) vaccines by at least 50%. Key stakeholders, mainly caregivers, community health workers, healthcare providers, parents, and religious and community leaders, were actively engaged throughout the HCD process, ensuring that solutions were developed collaboratively with the direct beneficiaries.

The rest of this paper is organized as follows: the materials and methods section outline the research design, including co-creation workshops, data collection, analysis, challenge definition, solution design, and implementation methods. The results section presents findings on barriers to routine immunization, key opportunities for generating innovative solutions to reduce the zero-dose gap in Ilala, interventions, prototyped solutions, and their implementation. The discussion interprets these findings about the research objectives and relevant literature. Finally, the conclusion summarizes key insights, offers suggestions for improving digital systems, and highlights directions for future research.

## 2. Materials and Methods

### 2.1. Study Approach and Methodology

The study employed a human-centered design (HCD) approach, where participants in community-based co-creation workshops identified, prioritized, and implemented community-centric interventions to increase vaccination demand and address the zero-dose gap among under-five children in Tanzania. This approach leverages creativity and empathy to design solutions that are not only tailored to the community’s specific needs but also foster active engagement and collaboration among stakeholders at all levels [19]. The study used persona models with fictional characters to represent key actors’ immunization needs, values, aspirations, abilities, and limitations. Initially, five personas were considered: caregivers, community health workers (CHWs), healthcare workers (HCWs), religious leaders, and community leaders. However, rapid inquiry in Ilala revealed that mothers are often the breadwinners, with grandmothers assuming caregiving roles.

Additionally, men and traditional healers were identified as key stakeholders in the decision-making process related to immunization. As a result, the caregiver persona was adapted to include both female caregivers (mothers) and grandmothers, and additional personas for men and traditional healers were created. We followed the UNICEF Journey to Health and Immunization framework (2020), which takes into account six stages in the health journey, helping us identify both barriers and enablers for key stakeholders (personas) in taking and sustaining health and immunization-related actions [26].

### 2.2. Study Population and Sampling

The study was conducted in Ilala District, Dar es Salaam, Tanzania, from March to December 2023. Ilala was selected due to its high number of zero-dose and under-vaccinated children [25]. Drawing on their extensive experience in health promotion and vaccine delivery, the Ministry of Health (MoH) played a significant role in identifying key stakeholders for participation in various phases of the HCD process, which engaged 483 diverse participants. The involvement of these stakeholders brought their unique perspectives and valuable experiences to the co-creation workshops and community engagements, which is essential for designing solutions tailored to the community’s needs and preferences. Participant involvement varied across the study’s stages based on individual expertise and availability. Given the dual objectives of developing community-centered interventions and building stakeholder capacity in HCD, 24 participants were consistently engaged throughout all HCD phases. This ensured they acquired the necessary skills to implement the co-created interventions and sustain vaccine uptake efforts using participatory approaches within their communities, workplaces, and healthcare settings. This approach aligns with UNICEF’s recommendation to integrate HCD into sub-national immunization coverage assessments to yield more impactful and sustainable initiatives [27]. A key innovation of this study was conducting multiple rounds of testing of the co-created interventions to challenge assumptions, collect feedback from communities, and uncover more user needs [28].

### 2.3. Co-Creation Workshops

The Ministry of Health defined the background and scope of the challenge, focusing on increasing support for and uptake of routine vaccinations, with the specific goal of reducing the number of children with zero doses of DTP and MR vaccines by at least 50%. To address this, two one-week co-creation workshops were conducted, where immunization stakeholders in Ilala were brought together to understand the immunization challenge, conduct rapid inquiry research, analyze and synthesize research findings, generate ideas, build prototypes, and engage communities to test and prioritize the prototypes iteratively. Over three months, three weeks were dedicated to planning and implementing one of the prototypes deemed most desirable by the community.

#### 2.3.1. Data Collection

In the HCD process, data are collected throughout all phases: problem diagnosis, solution design, and implementation. This process offers opportunities to uncover additional user needs, gather diverse perspectives, and identify hidden challenges. The study employed various data collection methods and tools through the co-creation workshops, including persona models, journey maps, rapid inquiry, brainstorming, discussions, prototyping, feedback grids, and prototype testing interviews.

#### 2.3.2. Challenge Definition and Diagnosis

In the first co-creation workshop, 45 immunization stakeholders in Ilala participated in the HCD training workshop to understand the challenge from the perspectives of the key actors through persona models and capture the multi-dimensional nature of the challenge through journey maps. Personas were developed using worksheets where participants detailed each persona’s motivations, frustrations, perceptions, routines, and influences. Challenge definition aims to understand users’ needs, identify assumptions and gaps in existing data, and facilitate diagnosis through rapid inquiry research.

Rapid inquiry employs interactive research methods to quickly explore the social, cultural, political, and economic influences and motivations behind people’s behaviors within a community. Participants used a rapid inquiry plan to explore areas of inquiry and ensure the right questions were directed to the right persona groups. Using discussion guides, participants prepared a list of questions and activities to be used during rapid inquiry. A discussion guide includes reminders of background information to share, open-ended questions to ask, and activities to administer. Sample activity included a worksheet that documents the day in the life of a given interviewee. At each point of the day, pain points and opportunities were identified. Participants visited health facilities and the Ilala community to conduct rapid inquiries to healthcare workers, community health workers, community leaders, and caregivers. Participants unpacked findings from the rapid inquiry using a Miro board and an interview highlights worksheet, capturing notes, quotes, and surprising moments. This was followed by analysis and synthesis of the unstructured data using open discussions and a 5-whys worksheet to uncover barriers, root causes, and opportunities for designing solutions.

#### 2.3.3. Solution Design

The design of interventions using HCD approaches is achieved through idea generation, prototyping, and iterative testing. During brainstorming sessions, participants leveraged identified opportunities to generate 309 ideas to address barriers to routine immunization. These ideas were then prioritized based on desirability, viability, and feasibility metrics, narrowing them down to 49 ideas suitable for prototyping.

In the second co-creation workshop, 31 participants participated in HCD training to further prioritize ideas, create prototypes, and iteratively test and refine these prototypes. Using a participatory approach and a live poll, the impact–effort matrix was used to score and rank 49 ideas based on impact and feasibility. The top six ranked ideas were carried forward into the prototyping phase. Participants then reviewed these ideas to identify patterns and themes by comparing them with the remaining ideas.

Prototyping is the process of making ideas tangible. Prototypes are an early, quick, and cheap mechanism to test key assumptions and assess if interventions meet user needs, providing room for feedback and improvements [29]. Participants used idea dashboards to describe each of the six ideas and state how they work by capturing critical functionalities. This process was followed by creating six paper prototypes for each idea. Community members tested the prototypes in Ilala. Feedback was unpacked on feedback grids to capture the features that users liked or did not like about the prototype, the questions asked, and new ideas that were suggested and worth trying. Participants analyzed the feedback on the grids and used it to iterate the prototypes by rebuilding them to incorporate new recommendations and remove some features. After 3 iterations of prototyping and testing through community engagements, the three most liked and desirable prototypes by the community of Ilala were selected for implementation.

#### 2.3.4. Theory of Change

The Theory of Change (ToC) framework [30] was applied to integrate the learnings from the HCD process, guide the implementation plan, and track the progress of interventions over time. The ToC mapped the system of stakeholders, activities, outputs, and outcomes that were anticipated to lead to the long-term goal of reducing the number of children with zero doses of Diphtheria–Tetanus–Pertussis (DTP) and Measles–Rubella (MR) vaccines by at least 50%. The ToC facilitated the development of a shared strategy for change and laid out a clear plan for monitoring and evaluating progress, allowing the study to test assumptions and track both intermediate and long-term outcomes against the goal of reducing zero-dose cases.

#### 2.3.5. Solution Implementation

The contribution of this study lies in applying HCD to the implementation phase of one of the three prioritized ideas. Existing literature indicates that the use of HCD beyond the design phase is still in its early stages, with most HCD studies not covering the entire project cycle [31,32].

The MoH, in collaboration with key stakeholders and representatives of participants who had undergone HCD training until the design phase, collaboratively established the implementation plan for the vaccine advocacy idea. The vaccine advocacy idea lies in using Ilala district leaders to advocate for vaccine uptake to community leaders and members. The team reviewed Ilala immunization health facility data to establish a five-year trend analysis for DPT3 and MR2 to identify low-performing areas needing prioritization during implementation. Through planning meetings and consultations, the team designed implementation tools and activities. The team set daily implementation targets for implementers to achieve and established criteria to guide the implementation duration for each ward. Key elements established during implementation planning include the number of advocacy stages, monitoring process, control mechanism, and reporting mechanism. Due to community engagements and open discussions, the advocacy prototype underwent another iteration to incorporate additional user needs and views.

During implementation, the District Commissioner (DC) advocated to 430 people in 2 different phases. In phase 1, the DC gathered council leaders to advocate for the sustainable and impactful implementation of vaccine mobilization and sensitization in their areas in the Ilala district. In phase 2, the DC, with other district leaders, gathered community leaders, HCWs, and CHWs in two low-performing constituencies to advocate for vaccine uptake. The DC shared the implementation target, where 56.5% of families in Ilala were expected to be reached (259,200 families). The DC provided two lists of wards: one where house-to-house mobilization would be completed in one week and another where it would extend over two weeks. Community-level advocacy presented an opportunity to gather feedback directly from the community leaders on the following key issues: understanding of HCD, opinions and views on the advocacy idea, and additional inputs to strengthen the implementation of HCD ideas.

House-to-house mobilization and sensitization commenced four days after completing the advocacy at the community level. In total, 189 CHWs were engaged to reach families in 26 wards in Ilala. Each CHW was accompanied by an HCW responsible for administering the vaccine. The CHW and HCW pair filled in vaccination data using a reporting tool designed based on specific indicators. The data were verified by the Street chairperson and submitted to the ward level for aggregation. The Ward Health Officers supervised the vaccination exercise in their respective wards. Ward-level vaccination data were verified and aggregated at the council level.

## 3. Results

### 3.1. Demographic Information

As shown in Table 1, the research involved a diverse group of 483 participants, with 24 consistently engaged throughout all phases of the human-centered design (HCD) process. Table 2 presents a list of participants involved in testing ideas and prototypes. Such stakeholders were categorized into various groups: researchers from higher learning institutions (HLIs); national government officials, mainly from the Ministry of Health (MoH); regional and district government leaders such as the Regional Medical Officer (RMO), District Executive Director (DED), District Commissioner (DC), mayor, and the Council Health Management Team (CHMT); local government leaders such as ward executive officers (WEOs), Mtaa executive officers (MEOs), Mtaa chairpersons, environmental health officers, and members of local government councils; civil society organizations such as the Shivyawata–Tanzania Federation of Disabled People’s Organizations, the TADIO network of community radio stations in Tanzania, the Chavita (association of people with hearing impairment), and BBC Media Action; community health workers (CHWs) and caregivers; and international organizations such as UNICEF, WHO, JHPIEGO, and AMREF.

### 3.2. Identified Barriers and Key Opportunities

Based on a rapid inquiry and observation, multiple barriers to routine immunization, as depicted in Table 3, were identified from various perspectives, including those of fathers, female caregivers, community health workers (CHWs), religious leaders, healthcare workers (HCWs), community leaders, and grandmothers. Each group faces unique challenges that impact the delivery and uptake of vaccinations.

From these identified barriers, nine key opportunities emerged, as shown in Figure 1, that were leveraged to generate innovative solutions to reduce the zero-dose gap in Ilala. These opportunities provided a foundation for targeted interventions to address each stakeholder group’s specific needs and concerns, ultimately increasing immunization coverage in the region.

### 3.3. Identified Interventions

From five opportunities, 309 ideas were generated through a collaborative brainstorming process involving diverse stakeholders, as summarized in Table 4.

### 3.4. Prioritized Interventions

Of 309 ideas generated, 49 were prioritized after thorough discussion and broad thinking. The criteria used for prioritization were desirability, viability, and feasibility. These 49 ideas were then ranked through a participatory approach, with each idea being reviewed and discussed in detail. From this ranking process, the top six ideas listed in Table 5 were selected for further exploration.

During the discussions, participants highlighted key issues and questions, pointing out barriers and opportunities that could be explored further. One notable opportunity is the government’s directive for health facilities to allocate resources through Health System Financing to enhance the capacity of CHWs. This initiative presents a valuable opportunity to strengthen CHW support. However, challenges remain, such as traditional healers’ reluctance to partner with healthcare providers, suggesting that their views and needs should be incorporated into the design process. Additionally, there were concerns about the effectiveness of initiatives to invite fathers and make clinics more male-friendly, with participants questioning the value of such efforts if the vaccination environment remains unwelcoming. Finally, the current guidelines prioritize men who bring their children for vaccination, highlighting a potential area for further development and reinforcement.

### 3.5. Prototypes

Prototypes of the top six ideas were created in different forms, not limited to sketches or storyboards on rough papers or even role play scenarios. These prototyped ideas were iteratively tested (three iterations). Feedback from the community was collected after each iteration, allowing the ideas to be refined and improved. As a result, three (3) of the most desirable and impactful ideas emerged, shaped by the inputs and suggestions from the community. The three ideas that emerged are summarized in Table 6. It was also observed that community members preferred and emphasized the involvement of CHWs in all the ideas. Therefore, the idea of the capacitation of CHWs was merged into the three selected ideas, and new versions for each idea were created. Three rounds of iterative testing were conducted on the refined versions of the three chosen ideas, with feedback from each round used to further enhance the prototypes up to version 3.

### 3.6. Prototype Implementation

The selected idea for implementation was advocacy through community leaders and community health workers (CHWs). Preparations for this implementation included creating a human-centered advocacy implementation plan involving collaboration with various stakeholders. The advocacy prototype was refined during this process based on feedback from the planning phase.

The refined prototype, as shown in Figure 2, underwent several modifications. Firstly, it broadened stakeholder inclusion by integrating previously omitted council leaders, such as the District Commissioner (DC), the Mayor, and the District Administrative Secretary (DAS). Secondly, the prototype streamlined the advocacy process by condensing the original four stages into two while still ensuring the participation of all relevant advocacy groups. The focus of the original prototype involved the DC advocating to ward-level leaders and Mtaa leaders (local/street leaders), who subsequently advocated to CHWs. However, since it was discovered during design research that CHWs maintain a closer relationship with the community, a more impactful approach would be for the DC to directly advocate for CHWs alongside ward leaders and Mtaa leaders (local/street leaders). Therefore, the advocacy flow was revised: the DC was now to advocate directly to council and community leaders. Community-level advocacy was restructured to incorporate the involvement of community health workers (CHWs), ward-level leaders, and street-level leaders.

During council-level and community-level advocacy, the DC championed vaccination to 430 participants, including government officials, regional leaders, district leaders, heads of departments, CHMTs, community leaders, CHWs, HCWs, media officers, and journalists.

House-to-house mobilization and sensitization commenced immediately after the advocacy meetings in 26 wards in the Ukonga and Segerea Constituencies. Depending on the established criteria, the house-to-house mobilization process spanned five or ten days per ward. Each of the 189 CHWs paired with an HCW was assigned a daily house-to-house target of 40 households, assuming an average of 4 families per household and 160 families per day. Ward Health Officers supervised house-to-house mobilization exercises, verified daily vaccination data, and confirmed visits with photos of house numbers or postcodes. Special tools were created to gather and organize vaccination data at community, ward, and council levels. The tool was designed to record the number of families reached, zero-dose cases addressed, defaulter cases resolved, and the total count of children in each family and household. As a control measure, the Mtaa chairperson verified and stamped the daily data report for each CHW. CHWs were to document visited families.

The data collection tool used during the house-to-house vaccine mobilization exercise also included fields to capture challenges, achievements, and views from CHWs, HCWs, community leaders, and community members. The data collected were analyzed, and it was observed that the challenges, successes, and opinions were related to the following thematic areas: RCH Card, Infrastructure, Tools, Community Participation, Resources, Availability of Children, and Vaccine Education. Table 7 summarizes the key challenges frequently highlighted by the participants. Most of the feedback from community leaders and families was positive, highlighting that most families were cooperative and loved empowering CHWs to mobilize families alongside HCWs. A notable recommendation was that prior notifications should be provided through PAs, radios, and TV.

In the planning phase, it was initially estimated that, on average, four families resided in each house within the Ilala area. However, subsequent observations revealed a variance in the ratio of families to houses across different divisions. Specifically, in Ukonga, the ratio was two families per house, whereas in Segerea, the ratio was three families per house. Outliers may have influenced the estimated ratio of families per house, as some houses in Ilala were found to accommodate huge families, with numbers reaching up to 16 families per house. During the exercise, 67,145 houses were visited (104% of the goal) as summarized in Table 8 and shown in Figure 3. Within these visited houses, 156,995 families were recorded, of which 131,088 (83%) were offered vaccination services.

Table 2 shows the number of zero doses and 1753 defaulters. However, as depicted in Figure 3, fewer zero doses (387) were identified during the vaccination exercise, while a significantly higher number of defaulters (9899) were identified. While the results may be alarming, they align with the HCD problem diagnosis and research findings. These findings revealed that Ilala, the central urban hub in Dar es Salaam, attracts frequent family movements. Documenting children as defaulters may not accurately reflect their vaccination status, as there is a high likelihood that they moved and received vaccination services in HFs other than the ones they were born in. The frequent movements within and outside of Ilala highlight a pressing challenge: the need for a mechanism to track the vaccination process and synchronize vaccination records within Ilala.

### 3.7. Identified Theory of Change

The theory of change aims to map out the system of people and activities that are assumed to collectively bring about the desired change. It provides a framework for collaboratively monitoring progress over time, ensuring that each step taken contributes to long-term goals. By tracking intermediate outcomes along the way, the approach ensures that all stakeholders are aligned and working towards common objectives. Table 9 depicts the theory of change for each persona built from identified barriers and co-created ideas. For instance, to have increased vaccine coverage in Ilala by 50%, CHWs need to be empowered to take on responsibilities such as administering vaccines, keeping records, and weighing babies. As a result, the workload for HCWs is significantly reduced. This leads to improved work–life balance for HCWs, resulting in a more positive clinic environment and the potential to extend clinic hours. As a result, families are more likely to attend and support regular healthcare services, leading to increased trust and accessibility.

## 4. Discussion

The use of HCD as a framework of this study helped involve and engage local community leaders, health workers, community health workers, and local communities of the study area to identify community-specific impediments to vaccination of under-fives and determine the possible solutions to address the issue. This design emphasizes the need for vaccination interventions to involve the target population in the ideation, prototyping, and implementation stages. The study established the root causes of under-fives defaulting and the zero-dose gap. Among the challenges identified were the inadequate support from local community leaders in sensitizing and monitoring under-five vaccinations, poor infrastructure (especially roads) to newly settled areas during the rainy season, parental hesitancy and unwillingness, limited involvement of male spouses, the absence of house numbers, and the lack of an effective surveillance and notification system. These challenges align with findings from Vasudevan et al. [10], Shearer et al. [26], Hogan and Gupta [33] and Reñosa et al. [34].

Efforts to bridge the zero-dose gap among under-fives in Tanzania and the rest of the Sub-Saharan countries of Africa highly depend on the availability and proper record-keeping of vaccination data [25]. As evidenced by this study’s findings, there needed to be more official data on unvaccinated children held by official designated authorities and those gathered in the study area. For example, the existing official vaccination data of the Ilala district showed that zero-dose vaccination is the most persistent vaccination challenge among the under-fives in the district, but this study found the opposite. As summarized in Figure 3, the study results suggest that vaccine defaulting among under-fives is the prevailing problem. According to the results, the district had a higher number of vaccine defaulters compared to the number of unvaccinated children. Consequently, the prevailing data discrepancies may lead to duplication of efforts and wrong demand-creation interventions.

The mismatch between existing official vaccination data and the field data collected in this study can be attributed to several factors. One major issue is the lack of an effective tracking and notification system for identifying vaccine defaulters, recording routine vaccination data, and sending reminders to parents and guardians. The current system relies on printed cards where healthcare workers manually record vaccination details, which are then kept by parents or guardians. This system poses challenges: cards can be lost, and parents may forget to check or remember vaccination dates. Additionally, some healthcare workers fail to record subsequent vaccination dates accurately. Consequently, this results in vaccine defaulting among under-fives. These findings align with existing research, which links parents’ forgetfulness and inadequate knowledge of vaccination schedules to incomplete vaccinations [35,36,37]. Similarly, the lack of vaccination cards has been linked to non-adherence to immunization schedules [38].

Moreover, families migrating into the Ilala district and seeking better maternity services have led to incomplete vaccination records when they return to their regions of origin after the initial doses of vaccines such as DTP1 and BCG. This migration has contributed to a rise in vaccination rates for some vaccines, like DTP and BCG, while rates for others, such as MR2, have declined. Additionally, the rapid expansion of new settlements in the district, characterized by high population density, fertility rates, and limited access to health services, exacerbates vaccine defaulting. Due to transport challenges, mobile vaccination services, such as Huduma Mkoba, are infrequent in these areas.

The rapid urbanization and expansion of new settlements contribute to persistent vaccine defaulting and zero-dose cases among under-fives. These findings are consistent with Sambayuka et al. [25], who identified rapid urbanization in Dar es Salaam, particularly in the Ilala district, as a factor in persistent zero-dose and under-vaccination issues. The study also highlighted that the under-capacitation of healthcare workers impacts the delivery of vaccination counselling and contributes to vaccine hesitancy and defaulting.

Notably, the achievement of targets, as shown in Figure 3, confirms that face-to-face interpersonal communication through house-to-house vaccination advocacy effectively addresses the zero-dose gap and reaches vaccine defaulters. This approach facilitates the accurate capture of vaccination data and mitigates hesitancy and negative influences often associated with mass vaccination campaigns.

Generally, the HCD framework effectively captured the quantitative house-to-house vaccine mobilization data and qualitative data on the challenges and achievements. It guided the determination of community-specific solutions to bridge the zero-dose gap and persistent vaccination defaulting among the under-fives in the study area. This study observed that the majority of participants (council and local community leaders, health workers, community health workers, and grassroots communities) held positive views of the HCD framework for employing community-based solutions to address vaccination challenges among under-fives and bridge the zero-dose and vaccination defaulting gaps in the study area. This framework can be pre-tested in other districts of Tanzania to determine its efficacy in eradicating the country’s zero-dose gap and under-vaccination of under-fives.

### 4.1. Lessons Learnt

The study demonstrated the transformative impact of engaging community members in every stage of the design and implementation process via co-creation workshops to address their challenges. Data-driven strategic advocacy and the involvement of trusted community members and leaders led to significant outcomes, including providing vaccination services to 131,088 families (83%). This effort reached 387 children with zero doses out of 2550 and identified many defaulters (9899 out of 1753). The success of this approach demonstrated the potential benefits of applying HCD innovation to public health issues, ensuring that the interventions were grounded in the realities and needs of the target population. The implementation team recommended extending this model to different regions and globally and integrating it with training institutions to ensure formalization and local sustainability.

### 4.2. Limitations of the Study

The study provides valuable insights into using an HCD approach to address vaccine hesitancy and increase uptake among under-five children. However, it was conducted in a single district. The Ilala district may not represent the entire country, limiting the generalizability of the findings to other regions with different cultural, economic, and healthcare dynamics. The insights and interventions developed in Ilala may not directly apply to rural or more socio-economically disadvantaged regions, which might face unique challenges in vaccination uptake. Additionally, the study focused primarily on stakeholders involved in child immunization, such as caregivers, CHWs, and HCWs. However, broader systemic challenges, such as supply chain issues, vaccine storage, and distribution, were not deeply examined. These factors could influence vaccine availability and accessibility, impacting the overall success of the interventions.

### 4.3. Recommendations

Future research should expand on these findings by conducting similar studies in diverse settings and incorporating broader factors influencing immunization coverage. The Ministry of Health and relevant stakeholders should institutionalize HCD as a core component of the immunization strategy. HCD should be used beyond immunization efforts. Scaling the use of HCD across regions will allow for identifying more nuanced community-specific challenges and co-creating tailored solutions to ensure formalization and local sustainability.

## 5. Conclusions

Reducing the zero-dose gap and the number of under-vaccinated children in Tanzania, a country with complex socioeconomic dynamics, requires an innovative, ethical, and practical approach like human-centered design (HCD). HCD helps overcome community hesitancy and misconceptions about vaccination by addressing the unique needs of each community and co-designing relevant, tailored interventions. The house-to-house campaigns implemented as part of the proposed community centric interventions raised awareness, increased vaccine uptake, and strengthened the community’s role in health advocacy. This demonstrates that applying HCD principles in public health interventions can effectively tackle challenges like zero-dose vaccination while fostering community engagement and sustainable change.

## Figures and Tables

**Figure 1 vaccines-13-00038-f001:**
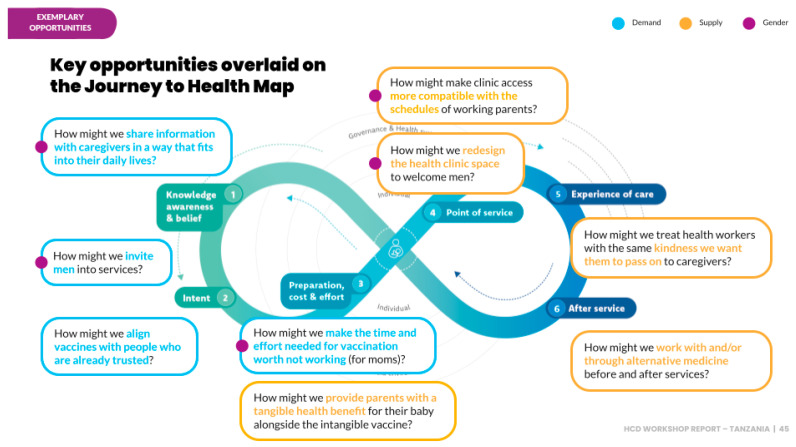
Key opportunities mapped onto the journey to health map. Source: HCD Training Report in Ilala (2023).

**Figure 2 vaccines-13-00038-f002:**
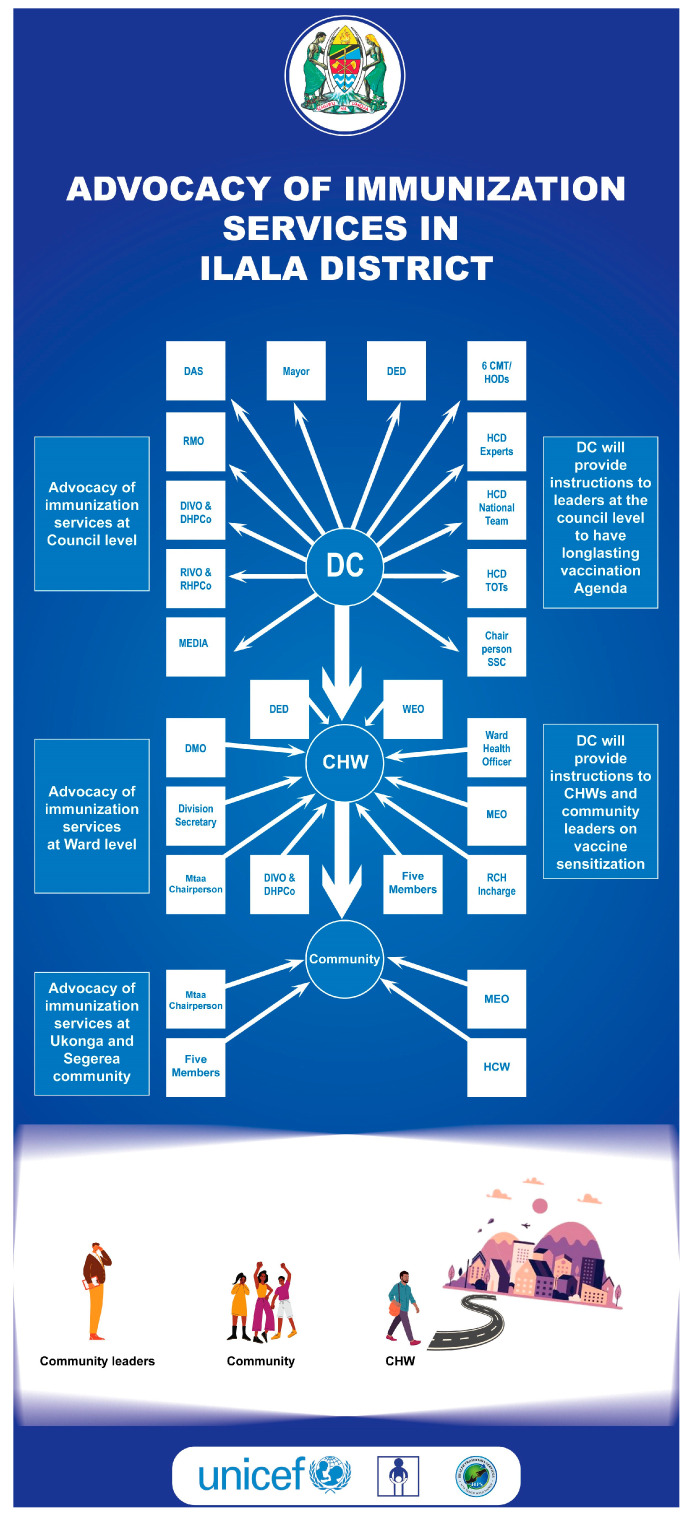
Refined prototype for the advocacy idea.

**Figure 3 vaccines-13-00038-f003:**
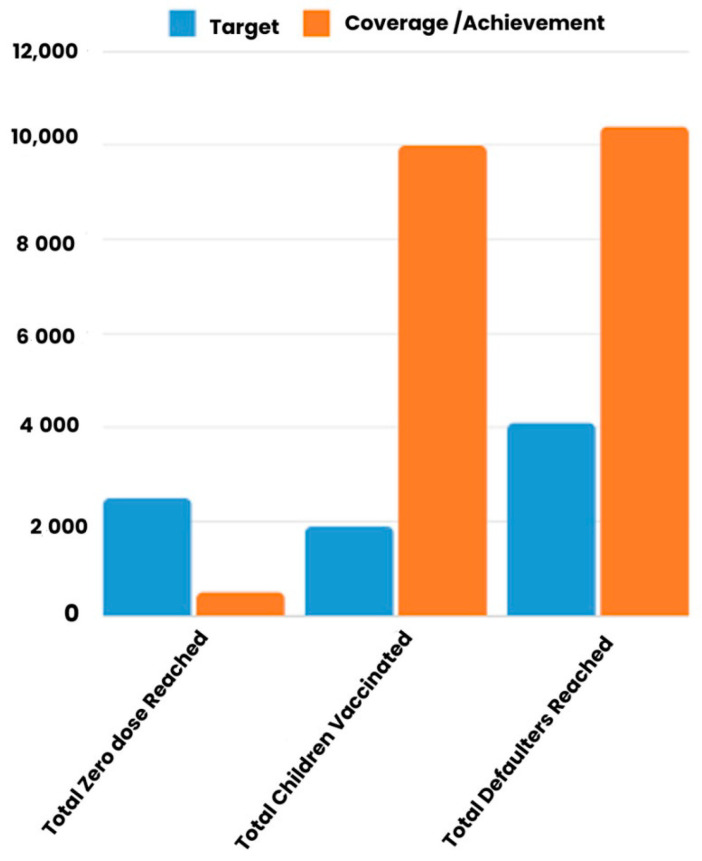
House-to-house sensitization and mobilization data.

**Table 1 vaccines-13-00038-t001:** Human-centered design (HCD) workshops participants list.

Participant Group	Number of Participants		
	HCD Training	Prototyping and Iterative Testing	Idea Implementation
Researchers from Higher Learning Institutions (HLIs)	12	7	7
National Government Officials from The Ministry of Health (MoH)	5	4	7
Regional and District Government Leaders (Regional Medical Officer (RMO), District Executive Director (DED), District Commissioner (DC), Mayor and Council Health Management Team (CHMT))	2	7	30
Local Government Leaders (Ward Executive Officer (WEO), Mtaa Executive Officer (MEO)/Local Executive Officer, Mtaa Chairpersons (Local/Street Chairperson), Environmental Health Officers, Members of Local Government Councils)	1	1	137
Civil Society (Tanzania Association of the Deafblind (TADIO), Tanzania Federation of Disabled People’s Organizations (Shivyawata), Tanzania Association of the Deaf (Chavita), Media)	16	3	23
Community Health Workers	2	2	178
Health Care Workers	3	1	46
International Organizations (United Nations International Children’s Emergency Fund (UNICEF), World Health Organization (WHO), Johns Hopkins Program for International Education in Gynecology and Obstetrics (JHPIEGO), and African Medical and Research Foundation (AMREF))	4	6	3
**TOTAL**	**45**	**31**	**431**

**Table 2 vaccines-13-00038-t002:** Human-centered design (HCD) iterative testing participants list.

	Number of Testers		
Participant Group	Idea Testing-Round 1	Prototype Testing-Round 2	Prototype Testing-Round 3
Caregivers	2	4	2
Regional and District Government Leaders (RMO, DED, DC, Mayor, and CHMTs)		6	2
Local Government Leaders (Community Leaders)	1	2	2
Community Health Workers	1	8	2
Health Care Workers	1	8	2
Religious Leaders	1	2	
**TOTAL**	**6**	**30**	**10**

**Table 3 vaccines-13-00038-t003:** Identified barriers for different persona groups.

S/N	Persona	Barriers
1.	Father	- There is the normalization of gender roles. - Fathers have been left out of child health initiatives, primarily targeting mothers. - Limited sources of health information reaching fathers. - The community prefers messages to go to mothers and fathers rather than fathers alone. - They questioned whether the father could take care of the baby alone.
2.	Female Caregivers	- Low motivation to skip earning to take children to clinics accounting for distance, long - wait times, and vaccine stockouts. - Many caregivers have no time or access to technology or social media, limiting their access to health information. - Claims that vaccine treatment shifts based on who is in the system and who is not; a very different experience for those who are not connected. - Many feel that traditional medicines are cheaper, trusted, and more accessible to them and would rather wait and treat their child traditionally if they fall ill.
3.	Community Mobilizers (CHWs)	- There is conflicting information about the role of the mobilizers and what they are doing for the community vs. the government, which impacts their influence. - Erratic incentives for mobilizers undermine their reputation; society thinks they are being paid, but it is an unpaid position. - Many in society feel that community mobilizers are paid and care more about the money than the community’s health. - CHWs must complete a heavy amount of data reporting, which takes time and makes it hard to do part-time work. - Health equipment (masks, protective equipment) makes the community think of an outbreak, not safety.
4.	Religious Leaders	- The leader sees and hears about wrong vaccination practices in the community but does not have the correct information to correct it. - Religious leaders dismiss parents who did not take their children to be vaccinated in their interactions, which can cause echo chambers. - While higher-level religious leaders have formal training for immunization, local religious leaders are teaching themselves.
5.	Health Care Workers (HCWs)	- Data reporting and logbook requirements are very time-consuming. - Vaccination centers are crowded with congested corridors, and workers are tired of keeping up with the influx of patients. - Patient feedback is not part of the job description, along with no employee feedback system. - HCWs work as they are trained. Updates are given through guidebooks but not personal guidance. - HCWs work in poor working conditions without incentives or overtime pay: “She ignores what she sees so she can do her work.” - Since COVID-19, there has been an increase in the question of the importance of routine vaccination.
6.	Community Leaders	- Some community leaders have their attention divided between their constituents and professional clients. They are constantly receiving phone calls and plugged into helping the community. - Community leaders are also caregivers for elderly parents or children, which reduces their office days and hours. - Community leaders focus on social issues that matter to the community. They do not see health issues as social issues and, therefore, do not see child vaccination and do not consider it to be a part of their responsibilities.
7.	Grandmothers	- Some grandmothers are breadwinners and work odd hours to earn money and take care of the home. - They feel cut off and outside of the conversation around vaccination and immunization. - Grandmothers know about health and use traditional medicine practices and healing. - They are proactive about vaccinations, but they hear rumors such as a person dying a day after the vaccine, and these stories become stuck in people’s minds.

**Table 4 vaccines-13-00038-t004:** Number of ideas generated for each design challenge (opportunity).

S/N	Persona	Number of Generated Ideas	Design Challenge
1.	Traditional Healer	80 ideas	How might we work with/through alternative medicine?
2.	Religious and Community Leaders	80 Ideas	How might we align vaccines with people who are already trusted
3.	Health Care Workers	68 Ideas	How might we treat healthcare workers with the same kindness we want them to pass on to the caregivers?
4.	Caregivers (Mothers, Grandmothers)	60 Ideas	How might we make her get her child vaccinated worth not working?
5.	Men	21 Ideas	How might we invite men into services?

**Table 5 vaccines-13-00038-t005:** List of prioritized interventions.

S/N	Intervention
1.	Having health facilities notify and alert local leaders and Wajumbe (Local Government Representatives) about vaccination dates.
2.	Giving priority to fathers when they come to the hospital with kids.
3.	Improve CHWs’ capacity to assist nurses and work with them.
4.	Use parents, kids, and grownups who got vaccinated to influence others.
5.	Use local government leaders and house representatives for vaccine advocacy.
6.	Put vaccination ads in religious places and centers.

**Table 6 vaccines-13-00038-t006:** List of most desirable interventions.

S/N	Intervention
1.	Having health facilities notify and alert local leaders and Wajumbe (Local Government Representatives) about vaccination dates.
2.	Use parents, kids, and grownups who got vaccinated to influence others.
3.	Use local government leaders and house representatives for vaccine advocacy.

**Table 7 vaccines-13-00038-t007:** List of challenges.

S/N	Challenge	Frequency
1.	Poor infrastructure (roads) due to rainy season	17
2.	Parents refusing having their children vaccinated	12
3.	The absence of tools such as gumboots, raincoats, and umbrellas	9
4.	Houses do not have house numbers	9
5.	Children born at home are not taken to health facilities	5

**Table 8 vaccines-13-00038-t008:** Household-level vaccine mobilization results.

S/N	Indicators	Ilala Community Vaccine Mobilization (Ukonga and Segerea)		
		Target	Actual	(%)
1.	Households reached	64,800	67,145	104
2.	Number of families present in the visited households	259,200	156,995	61
3.	Number of families reached	156,995	131,088	83
4.	Zero doses reached	2550	387	15
5.	Defaulters reached	1753	9899	565
6.	Total children vaccinated	4303	10,286	239
7.	Eligible children who were missed		2127	
8.	Total children in families that were reached		90,158	

**Table 9 vaccines-13-00038-t009:** Theory of change.

IMPACT	1. Increased Coverage by 50% in Ilala		
OUTCOME	1.1 Families Support and Regularly Attend Services		
Inputs	Activities	Outputs	Intermediate Outcomes
HCWs who are overworked + CHWs who are trusted by the community	Evolve CHWs’ role to conduct routine vaccination activities like vaccination, record keeping, weighing babies	Less working hours for HCWs	Happier HCWs, better clinic experience, opportunity to open clinics for more hours
CHWs whose presence is critical to the community accepting services but who are believed to be motivated by money	Learned in prototyping: What helps CHWs gain trust and address misinformation? Incentives like transport support Tools like megaphones Recognition from community Recognition from health sector (including gov.)	CHWs have reliable, regular engagement with community (NOTE: sporadic engagement is causing rumors that they only work when paid)	Community trusts CHWs for advice and reminders
Caregivers who are not available when facilities are open + HCWs who do not have more time during the weekday	Offer services on Saturdays	More families receiving services	Women do not need to choose between making money and attending health services
**OUTCOME**	**1.2 Community ownership of vaccination and support of preventative medicine**		
**Inputs**	**Activities**	**Outputs**	**Intermediate Outcomes**
Grandmothers who have knowledge about vaccines + CHWs who are trusted by the community	Partner to provide outreach services to the community	Specific days where CHWs can reach out to grandmothers and provide vaccines	Vaccination is not only a mother’s responsibility
Religious leaders who currently have informal training + Implementers in care who are not really trained	Receive formal training (beyond chairperson) that can be applied into their career	Links to information with their focus (religion)	Enhanced confidence during religious sermons
Community leaders who are involved in social issues in the community, but do not see health issues as social issues	Receive a health/vaccination induction after they come to office + Become involved in mobile clinics that are close to their community	More attendance at clinics	Social norm in community is to trust in vaccines

## Data Availability

The original contributions presented in the study are included in the article, further inquiries can be directed to the corresponding author.

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
