# Peer review of "Using Human-Centered Design to Bridge Zero-Dose Vaccine Gap: A Case Study of Ilala District in Tanzania"

_vaccines, 2025, doi:10.3390/vaccines13010038_

Round 1

Reviewer 1 Report

Comments and Suggestions for Authors

First of all congratulations to the authors, all involved researchers and to the participants and community members for making this study possible with such interesting results. Hongera sana.

I have some remarks on some minor issues but these can be rectified quite easily, I suppose. Please see my comments below.

Provide an ethics statement as this study should have been approved by NIMR or a similar official body before commencement.

Provide some information on funding for this research. The reason behind my question ist that especially work time was needed very much for conducting this study. It should be clear that additional resources would be necessary to add this type of vaccination effort to the routine EPI programme.

Provide a reference for figure 1 which seems to be a copy of a figure in the official HCD report.

It may be useful to provide an English translation of figure 2 as a supplement. Otherwise the messages of the poster will not be understood by the majority of readers who are not fluent in Swahili.

The authors may consider changing figure 3 into a better design.

Page 13, section 3.7: is the theory of change not a new concept which should have been introduced under methods first? Or did I miss it in the methods section? At least provide a reference for this theory.

lines 416-417: the last sentence does not read well. Cite the authors' names instead of only the reference numbers.

lines 439-440: provide a reference for the last statement

lines 449-453: here the authors mention qualitative interviews with HCWs and CHWs. Did I overlook them in the results section? If not they need to provide a reference. Or do they mean that Gichogo et al is the reference? But this reference is not in the list.

The sections 4.1. Lessons learnt and 4.2. Recommendations are a bit short and not fully aligned with the rest of the paper. Perhaps the authors can think about an alternative wording.

Please check all the abbreviations if they are introduced in the text. Because of the many abbreviations used perhaps a list would be helpful.

Provide the English translation of the Swahili words which are used in the text. Not everyone would know the meaning of mtaa or wajumbe. Perhaps it would be best to mention the Swahili word first and then provide the English translation in brackets.

Comments on the Quality of English Language

correct some typos, sometimes a word is missing

Author Response

Comment 1: Provide an ethics statement as this study should have been approved by NIMR or a similar official body before commencement.

Response 1: This manuscript was generated and prepared from the routine implementation and sensitization of vaccine. Findings from the implementation brough something new and a different way to view and approach public health challenges. The observations/findings pushed us to share these with other World so as we can influence or help others on the same.

Comment 2: Provide some information on funding for this research. The reason behind my question is that especially work time was needed very much for conducting this study. It should be clear that additional resources would be necessary to add this type of vaccination effort to the routine EPI program.

Response 2: Yes, the implementation required extra time and resources. Thus, the initiative was financially supported by UNICEF through the HCD approach that aim to accelerate positive behaviour change on vaccine uptake aiming at combating zero dose and defaulters in Ilala District

Comment 3: Provide a reference for figure 1 which seems to be a copy of a figure in the official HCD report. 

Response 3: Added the source as: Source: HCD Training Report (2023)

Comment 4: It may be useful to provide an English translation of figure 2 as a supplement. Otherwise, the messages of the poster will not be understood by the majority of readers who are not fluent in Swahili. 

Response 4: The figure 2 has been translated to English. 

Comment 5: The authors may consider changing figure 3 into a better design.

Response 5: Replaced the figure with one that is more visible

Comment 6: Page 13, section 3.7: is the theory of change, not a new concept which should have been introduced under methods first? Or did I miss it in the methods section? At least provide a reference for this theory. 

Response 6: Section 2.3.4. on Theory of Change added in Materials and methods. Reference added. Section 3.7 revised to include a table of formulated theory of change for each persona, generated ideas, and outcomes that lead to the impact.

Comment 7: Lines 416-417: the last sentence does not read well. Cite the authors' names instead of only the reference numbers. 

Response 7: The sentence has been revised and cited accordingly,

Comment 8: Lines 439-440: provide a reference for the last statement 

Response 8: A reference has been added.

Comment 9: Lines 449-453: here, the authors mention qualitative interviews with HCWs and CHWs. Did I overlook them in the results section? If not, they need to provide a reference. Or do they mean that Gichogo et al. is the reference? But this reference is not in the list. 

Response 9: The section has been revised accordingly, and the relevant reference has been included.

Comment 10: The sections 4.1. Lessons learnt and 4.2. Recommendations are a bit short and not fully aligned with the rest of the paper. Perhaps the authors can think about an alternative wording. 

Response 10: The lessons learned and recommendations have been revised to ensure alignment with the rest of the paper.

Comment 11: Please check all the abbreviations if they are introduced in the text. Because of the many abbreviations used perhaps a list would be helpful. 

Response 11: Added a list of abbreviations at the end, after the references

Comment 12: Provide the English translation of the Swahili words which are used in the text. Not everyone would know the meaning of mtaa or wajumbe. Perhaps it would be best to mention the Swahili word first and then provide the English translation in brackets.

Response 12: The word ‘Mtaa’ is changed to local/street. Wajumbe - Local Government Representatives

Comment 13: Comments on the Quality of English Language correct some typos, sometimes a word is missing 

Response 13: Corrections were made to address typos, and missing words were added where necessary.

Reviewer 2 Report

Comments and Suggestions for Authors

Estimated Authors,

I've been invited to review the present study, entitled "Bridging Zero Dose Gap: Success Stories and Lessons Learnt in Applying Human-Centred Design in Ilala District, Tanzania".

Authors have reported on the very important initiative performed in the Ilala district by means of the application of HCD4H. This design is a problem-solving process that begins with understanding the human factors and context surrounding a challenge and has been recommended for being used where there exists a high degree of ambiguity along with a commitment to creating people-centered, high-value solutions to address complex problems (https://link.springer.com/article/10.1007/s11606-023-08500-0).

The Authors have properly documented and explained the reasons for applying HCD4H in this settings, and this information is available in initial sections of the paper.

Unfortunately, a main shortcoming of HCD4H is that while long-term outcomes (in this case, vaccination records) could be appreciated quite easily, in a shorter time period the actual efficacy and the cost-efficacy of HCD4H is quite more difficult to quantify, as for the present article. Therefore, the reduced numeber of eventual outcomes that have been reported should be acknowledged as a sort of "structural" limit of this approach rather than of the paper.

From my point of view, some minor improvements could be envisaged, and more precisely:

1) Table 1, 2: please solve all acronyms within the tables and the caption of the tables.

2) Figure 2 should be translated into English at least as a Supplementary material.

3) Figure 3: the subfigure on the left (i.e. the table) has a cell marked in green. Please explain and, where possible, redesign in accord to the Journal's guidelines.

4) The study lacks of two proper sections focusing on limits (ideally to be placed before the current section 4.2) and a final "conclusions" section would radically improve the proper understanding of the study.

Author Response

Comment 1: Table 1, 2: please solve all acronyms within the tables and the caption of the tables. 

Response 1: All acronyms in Table 1, Table 2, and their captions have been expanded.

Comment 2: Figure 2 should be translated into English at least as a Supplementary material. 

Response 2: The Figure 2 has been translated to English 

Comment 3: Figure 3: the subfigure on the left (i.e. the table) has a cell marked in green. Please explain and, where possible, redesign in accord to the Journal's guidelines. 

Response 3: The table has been redesigned according to the Journal’s guidelines and the cell that was marked in green has been unmarked

Comment 4: The study lacks of two proper sections focusing on limits (ideally to be placed before the current section 4.2) and a final "conclusions" section would radically improve the proper understanding of the study. 

Response 4: Two new sections, one addressing the study's limitations and another providing the conclusion, have been added.

Round 2

Reviewer 2 Report

Comments and Suggestions for Authors

Estimated Authors,

since its first version, the present paper has been extensively improved. Therefore, there is currently no need for further revisions or improvements.